# Reported Adverse Events in Patients with CF Receiving Treatment with Elexacaftor/Tezacaftor/Ivacaftor: 5 Years Observational Study

**DOI:** 10.3390/jcm14124335

**Published:** 2025-06-18

**Authors:** Francesca Lucca, Ilaria Meneghelli, Gloria Tridello, Francesca Buniotto, Giulia Cucchetto, Sonia Volpi, Emily Pintani, Valentino Bezzerri, Marco Cipolli

**Affiliations:** 1Cystic Fibrosis Center, Azienda Ospedaliera Universitaria Integrata, 37126 Verona, Italy; francesca.lucca@aovr.veneto.it (F.L.); ilaria.meneghelli@aovr.veneto.it (I.M.); gloria.tridello@aovr.veneto.it (G.T.); psico.fc@aovr.veneto.it (F.B.); giulia.cucchetto@aovr.veneto.it (G.C.); sonia.volpi@aovr.veneto.it (S.V.); emily.pintani@aovr.veneto.it (E.P.); v.bezzerri@unilink.it (V.B.); 2Department of Life Sciences, Health and Health Professions, Link Campus University, 00165 Rome, Italy

**Keywords:** cystic fibrosis, elexacaftor, tezacaftor, ivacaftor, CFTR modulators, drug safety

## Abstract

**Background:** Elexacaftor/tezacaftor/ivacaftor (ETI) treatment is showing remarkable beneficial effects in people with Cystic Fibrosis (pwCF) harboring the F508del mutation in the Cystic Fibrosis Transmembrane conductance Regulator (*CFTR*) gene. Although this therapy is generally well tolerated in pwCF, some adverse events (AEs) have been recently described both in controlled studies and in post-marketing observations. **Methods**: We followed 414 pwCF carrying F508del *CFTR* that initiated ETI treatment, recording AEs for a period of 5 years. **Results**: A total of 142 AEs were reported. The most frequent AEs in the whole cohort were liver marker elevation, skin rush, epigastric pain, headache, and depression. Considering pediatric subjects, psychiatric and gastrointestinal disorders were the most frequent AEs. Only one patient reported a severe AE, leading to treatment discontinuation. In case of AEs, different decisions on ETI treatment were made, including temporary interruption and temporary or permanent dosage modification. **Conclusions**: Throughout the long-term observational period, almost 21% of pwCF experienced at least one AE. Psychiatric disorders, in particular attention deficit, were the most prevalent issue in our pediatric cohort, whereas adult patients mainly reported depression, anxiety and sleep disorders. This study therefore strengthen the recommendation of screening for changes in mental health during ETI treatment. AEs led to the permanent reduction of ETI dosage in 32% of cases, raising the issue of safety in relation to dosage reduction, efficacy, and minimum ETI levels. Eventually, this study highlights the need for a longitudinal monitoring of ETI safety since a significant number of AEs occurred after one year of treatment.

## 1. Introduction

Cystic fibrosis (CF) is an autosomal recessive disorder caused by mutations in the CF transmembrane conductance regulator (*CFTR*) gene, encoding a chloride and bicarbonate channel essential for the maintenance of the correct salt and water balance at the surface of epithelial cells [1,2].

Although more than 2000 mutations in *CFTR* have been reported so far, F508del is the most frequent pathogenic variant. This mutation causes misfolding of CFTR protein, leading to a defect in trafficking from the endoplasmic reticulum (ER). Over the past decade, the introduction of CFTR modulator therapies has transformed the CF clinical landscape. The first of these to be developed was the CFTR potentiator ivacaftor, improving CFTR protein function in the apical membrane of epithelial cells [3]. Other CFTR correctors improving the trafficking of unfolded F508del CFTR protein to the cell surface were subsequently developed. Currently, triple-combination elexacaftor-tezacaftor-ivacaftor (ETI) treatment has proved to be highly effective and safe when administered in clinical trials to patients with CF (pwCF), aged 2 years and older, harboring at least one F508del *CFTR* allele [3,4,5,6,7,8]. Recently, the Food and Drug Administration (FDA) expanded its ETI approval to a total of 272 *CFTR* mutations, allowing more pwCF to access this treatment.

ETI significantly improved lung function in pwCF, reducing the pulmonary exacerbation rate and decreasing hospitalizations [4]. It also increased body mass index, improved quality of life, decreased the sweat chloride concentration, and reduced infections. In addition, ETI was generally well tolerated in trials [5,7,9,10,11,12,13,14]: real-world data corroborate these findings, affording a valuable source of evidence on ETI’s tolerability and safety profile [15]. However, some adverse events (AEs) have been described both in controlled studies and in clinical practice. The most commonly reported AEs were increased liver marker levels and headache [3,5,9,15]. Post-marketing observations have highlighted additional AEs that were previously either not documented or under-reported [16,17,18,19,20], including neuropsychiatric symptoms and dermatologic reactions that required dosage modification or discontinuation of the treatment.

The aim of this study is to report potentially drug-related AEs in a large cohort of pwCF receiving long-term ETI treatment, and to assess their consequences in terms of temporary or permanent drug regimen modification.

## 2. Materials and Methods

### 2.1. Study Design and Study Participants

Between October 2019 and December 2024, 414 pwCF regularly attending the Verona Cystic Fibrosis Center initiated ETI treatment. The study was approved by the Ethics Committee of the Azienda Ospedaliera Universitaria Integrata, Verona (approval number 433CET), and informed consent was obtained from all subjects.

Dosage regimens were prescribed in compliance with the package leaflet and Italian Medicines Agency (AIFA) requirements (patients older than 6 years, weighing <30 kg, to receive elexacaftor 100 mg once daily, tezacaftor 50 mg once daily, ivacaftor 75 mg every 12 h; patients weighing >30 kg to receive elexacaftor 200 mg once daily, tezacaftor 100 mg once daily, ivacaftor 150 mg every 12 h).

AEs were actively monitored by physicians during both outpatient and inpatient visits and via chart and record reviews throughout the observation period. Per our internal protocol, if a psychological disorder was suspected or patients with CF reported psychological symptoms, an interview with a CF psychologist was arranged. If deemed necessary, the patient was then referred to a child neuropsychiatric or psychiatric specialist. Pulmonary function were assessed quarterly, while a sweat chloride test was recorded annually; blood tests were done in accordance with AIFA guidelines (quarterly during the first year of treatment, then annually).

Percent predicted forced expiratory volume in 1 s (ppFEV1) was obtained by forced spirometry [21]. The parameter is expressed as the mean of measurements collected during consecutive 12-month periods (the year before ETI initiation, and every year of ETI treatment). Prevalence and type of AE were considered, regardless of the need for discontinuation of treatment. Serious AEs (SAEs) were defined in line with FDA and European Medicines Agency (EMA) criteria [22,23,24,25]. Identification of severe lung disease was based on ppFEV1 < 40 and/or ongoing evaluation for lung transplantation.

### 2.2. Statistical Analysis

Descriptive statistics were used to summarize patient characteristics: continuous variables were described by median and min-max values, whilst categorical variables were reported by absolute and percentage frequencies. Baseline characteristics were compared between patients presenting AEs events and those with none. Chi-square was used for categorical variables and, after having assessed the normality of the distribution for the continuous variables by the Shapiro-Wilk Test, the Mann-Whitney test was used, because the normality was not verified. The rate of AE occurrence during treatment was calculated, both for the overall period of ETI administration and for the first 30 days alone. Rates are reported as the number of AEs per 1000 days of treatment and were compared in relation to the main patient characteristics, using the Poisson exact test. A *p*-value < 0.05 was considered statistically significant. All analyses were done with the R software version 4.5 (R Foundation for Statistical Computing, Vienna, Austria (https://www.R-project.org, accessed on 26 May 2025).

## 3. Results

The evaluation comprised 414 patients older than 6 years (M 46%). One patient underwent lung transplantation after 3 months of treatment with ETI and was lost to follow-up. Patient characteristics are listed in Table 1.

Median duration of ETI treatment was 35.60 months [IQR 26.46–39.26]. During the period of observation, 142 AEs were reported, with 85 (20.53%) patients presenting at least one AE. Only one SAE was reported, consisting in epigastric pain which led to hospitalization and discontinuation of the drug. In the first two weeks of treatment, 47 AEs occurred (33.1%), while in the third and fourth weeks, and after 30 days, 15 (10.6%) and 72 (50.7%) AEs occurred respectively; in 8 AEs timing of presentation was not available (unknown).

We observed no differences in age, sex, severe respiratory impairment, or genotype between patients with and without AEs (Table 1). AEs distribution according to pwCF characteristics are shown in Table 2. The overall rate of AEs did not differ according to the main patient characteristics. However, when early events were considered, the AE rate was higher in pwCF with severe respiratory impairment, especially after 15 days of treatment (Table 2, parts B and C). The most frequent AEs were liver biomarker elevation (16.2%) and skin rash (11.27%), followed by epigastric pain (7.75%), headache (4.2%) and depression (4.2%). We harmonized AEs on the basis of System Organ Class (SOC) [19,20], categorizing anatomical or physiological systems involved: this ensured uniformity in reporting and analyzing data. The most frequent SOCs were psychiatric, investigations, skin and subcutaneous tissue disorders followed by gastrointestinal, musculoskeletal, and connective tissue disorders (Table 3). Characteristics of AEs differed between adult and pediatric pwCF. In pediatric pwCF, psychiatric disorders were the most frequent SOC, followed by skin and subcutaneous-tissue disorders, nervous-system disorders, investigations, and gastrointestinal disorders (Table 3). In adult pwCF, musculoskeletal and respiratory disorders were observed, whereas these were not reported in our pediatric cohort.

Overall, investigations, psychiatric, and musculoskeletal disorders occurred predominantly after the first two weeks of treatment, while skin and subcutaneous-tissue, gastrointestinal, and nervous-system disorders were reported mainly during the first two weeks.

Among the 24 psychiatric AEs as classified by SOC, 70.81% were first occurrences in the patients concerned. Different decisions regarding ETI treatment were made for the 142 AEs reported. Permanent discontinuation was rare (11% of all AEs), while temporary interruption, temporary dosage modification and permanent dosage modification were chosen in 17%, 25% and 32%, respectively (Figure 1). An internal protocol (see Appendix A) was applied for dose reduction by administering elexacaftor 200 mg, tezacaftor 100 mg, and ivacaftor 150 mg on alternating days with elexacaftor 100 mg, tezacaftor 50 mg, and ivacaftor 75 mg. Among the recorded AEs, no treatment change was necessary in 52 cases, permanent discontinuation was decided for 9 cases, and in 6 cases the initial dosage modification failed, also resulting in permanent discontinuation. Dose modification was successful in 44 cases and in 31 cases temporary adjustments resolved the AEs and the standard regimen was subsequently resumed.

Psychiatric disorders, investigation-related alterations, gastrointestinal, nervous system and musculoskeletal disorders most frequently resulted in permanent ETI dosage modification, whereas respiratory disorders, skin and subcutaneous tissue events mostly entailed only temporary dosage modification (Figure 1A, Table 4).

In the pediatric population, a more interventionist approach may be warranted, as a greater proportion of adverse events led to dosage adjustments or treatment interruptions. Moreover, while psychiatric and skin-and-subcutaneous-tissue disorders in adults most often did not necessitate changes in therapy, in children the majority of these reactions prompted either dosage modification or treatment alteration (Figure 1B,C).

## 4. Discussion

In this study we investigated potentially drug-related AEs in pwCF, treated with ETI in a real-life setting. Key strengths of this observational prospective study are that it encompasses a substantial patient cohort (414 subjects) and an extended follow-up period (median 35.6 months). A major limitation of the study is that it has been carried out in a single center; nevertheless, this also ensured consistency and reliability of the measurements. In addition, collecting AEs during clinic visits may be limiting, since some events could be missed if patients do not report them or physicians do not actively inquire, leading to an underestimation of AE incidence. In particular, mild events may be overlooked by patients, and their fear that reporting AEs could result in a reduction or discontinuation of ETI should also be considered. Our findings show that 20.53% of the treated cohort experienced at least one AE, whereas a SAE occurred in only one patient (0.25%). Only a few (11%) AEs led to permanent discontinuation of the study treatment (8 pwCF, i.e., 1.9% of the treated cohort). These figures indicate that ETI was quite well tolerated.

A recent meta-analysis of both clinical trials and real-world ETI studies [26] found an overall AE incidence of 27% to 98%, with SAEs accounting for 6.6%. The most common AEs were cough, rhinorrhea, headache, pulmonary exacerbations, nasopharyngitis, upper respiratory tract infections, and rash. The safety profile in adult and adolescent clinical trials [5,7,10,11] was generally consistent, with AE incidence reaching up to 98% in extension studies. Most AEs were mild or moderate in severity, with SAE rates of 4% [5] and 13.9% [7] in the initial short-term trials, rising to 30.4% [10] in the extension studies. The distribution of reaction types was similar in both the parent and extension studies. Most AEs were related to pulmonary exacerbations and CF symptoms, such as cough, increased sputum, nasal congestion, nasopharyngitis, and upper respiratory tract infection. Non-respiratory AEs were primarily headache, diarrhea, and fever. Elevations in creatine kinase, transaminases, and blood pressure were also observed. ETI discontinuation occurred in 1–3% of clinical-trial participants, most often due to elevated transaminases, hepatic complications, rash, or depression.

In the Phase 3 clinical trial on children (6–11 years) homozygous for F508del and a minimal-function mutation [12], 80% of those receiving ETI experienced AEs (mostly mild to moderate) and 6.7% had SAEs; the discontinuation rate was 1.7%. Headache, cough, and rash occurred in 30%, 23%, and 13% of children, respectively. A 96-week open-label extensions in pediatric populations mirrored these findings. Among children heterozygous for F508del and a minimal-function mutation, 98% experienced ≥1 AE and 10.8% ≥1 SAE [13], while in those with at least one F508del allele, the rates were 98.4% and 6.3%, respectively [14]. Most AEs were mild or moderate and consistent with CF manifestations or common childhood illnesses. Mall et al. [13] reported one discontinuation (0.8%) for steatorrhea. Wainwright et al. [14] reported aggressive behavior in one patient (1.6%). Generally, in these extension studies, the most frequent AEs were cough, headache, rhinosinus symptoms, pyrexia, upper respiratory infection, abdominal pain, constipation, and diarrhea. Rash and transaminase elevations occurred in about 10% and 5% of participants, respectively. Notably, ophthalmologic AEs were described, including cataracts in 2.5% and lenticular opacities in 0.8% of children [13], and one case of idiopathic intracranial hypertension with papilledema [14].

More recent real-world data have deepened our understanding of long-term ETI safety. French registry patients with severe pulmonary disease [15] experienced rash in 11%, gastrointestinal symptoms in 10.2%, myalgia in 4.7%, and headache in 4.2%. Three-fold increase of transaminases and two-fold increase in bilirubin relative to the upper limit of normal (ULN) appeared in 3.3% and 0.8%, respectively. Elevated creatine phosphokinase (CPK) was seen in 3.4% of cases. Rash led to temporary ETI interruption. Smaller series [27] reported AEs in 55% of severe patients, most commonly stomach ache (20%) and rash (15%). A Portuguese cohort in less-severe patients [28] highlighted headache (7.6%) and neuropsychiatric symptoms (12.6%). A FDA Adverse Event Reporting System (FAERS) pharmacovigilance analysis [29] confirmed labeled AEs but also flagged unexpected events, including anxiety, depression, insomnia, nephrolithiasis, and testicular pain, underscoring the need for cautious interpretation yet drawing attention to neurological and psychiatric signals not seen in extension trials. Pediatric real-world safety is generally reassuring. In a retrospective Italian cohort of 608 children on ETI [20], 18% experienced AEs (mostly urticaria/rash 41.2%, hypertransaminasemia 14.1%, headache 11.8%, epigastralgia 9.4%) and 2.7% SAEs. Persistent AEs included jaundice 40.8%, hypertransaminasemia 25.9%, rash 7.4%, and neuropsychiatric disorders 7.4%. Seven (1.1%) required definitive discontinuation due to severe hypertransaminasemia, persistent rash, depressive state, neuropsychiatric disorders, or diarrhea, whereas symptoms resolved in four. In Germany, Olivier et al. [30] described rash, liver-enzyme elevations, and CK increases. ETI dosing was adjusted in three patients for liver-enzyme elevations and resumed at a low dose after interruption in one child with idiopathic intracranial hypertension and papilledema.

We observed that 33% and 47% of AEs occurred within 2 and 4 weeks from ETI initiation, respectively. FDA data indicate that the median time of AE onset is 70 days, with the majority occurring within 4 weeks [29]. Early adverse events consisted primarily of skin and subcutaneous tissue disorders, gastrointestinal disorders, and nervous system disorders, whereas investigations, psychiatric disorders, and musculoskeletal disorders emerged predominantly after the first two weeks of treatment. Earlier AEs in the pediatric population (2- to 5-year-old and 6- to 11-year-old subgroups) were described using the FDA reporting system, and speculatively attributed to the higher metabolic rate in these age groups [31]. Here, we found that the most frequent AEs were liver function marker elevation, skin rash, epigastric pain, headache and low mood. The most frequently involved SOCs were investigations (mostly hepatic enzyme elevation), psychiatric, skin, gastrointestinal and musculoskeletal disorders. The prevalence of AEs did not differ in pediatric and adult pwCF, regardless of sex, severity of respiratory function impairment, or genotype, this trend being consistent with previously described pediatric data [24]. In our pediatric cohort of pwCF, psychiatric disorders were the most common AEs (35%), primarily presenting as concentration difficulties. In contrast, among adult patients, psychiatric AEs chiefly included low mood and depressive symptoms, anxiety, and sleep disturbances. A recent analysis of the UK Adverse Drug Reaction reporting system showed that the proportion of psychiatric AEs increased in the post-ETI period [32]. Zhang et al. retrospectively reviewed records of pwCF treated with ETI and found that, although mean annual PHQ-9 and GAD-7 scores remained unchanged, 22% of patients initiated or adjusted psychiatric medications. Additionally, 23% reported sleep disturbances following ETI initiation [33]. An Italian prospective assessment of neuropsychiatric symptoms [34] showed no change in anxiety questionnaire scores over time, but did document an increase in insomnia, and identified a higher prevalence of reported AEs in female patients (insomnia, headache, concentration problems, and brain fog).

Taken together, our data and previous reports suggest that systematic screening of psychiatric conditions needs to be taken into account. The high background prevalence of psychiatric disorders in pwCF has already been described. Although data from the US and German registries showed that patterns of depression prevalence did not change upon ETI treatment in adults [35], accurate evaluation of psychiatric symptoms through standardized questionnaires should be carried out for the pediatric population. Concerns may extend beyond psychiatric conditions, to encompass neurodevelopmental issues [36]. It should be borne in mind that screening for changes in mental health during the first 3 months of ETI is recommended in the European Cystic Fibrosis Society (ECFS) Standards of Care guidelines [37].

We found that AEs led to permanent dosage adjustments in 32% of cases; this raises the issue of safety in relation to dosage reduction, efficacy and minimum ETI levels. A recent case series [38] correlated AEs with ETI dosage, showing that dose reduction mitigated AEs while maintaining efficacy in terms of sweat chloride concentration. Nevertheless, the optimal dose adjustment strategy has not yet been identified and the long-term efficacy of different dosing strategies needs to be demonstrated.

In relation to permanent ETI treatment modification, whether by dosage adjustment or discontinuation, the causative AEs in most cases comprised liver marker elevation. This is not unexpected: one clinical trial found a higher incidence of liver involvement in comparison to placebo [5], albeit not resulting in the study treatment’s discontinuation. Longitudinal real-world data showed a non-significant increase in median ALT, AST, and total bilirubin values after 3 months of treatment [38,39].

## 5. Conclusions

Our study shows good tolerability for ETI, with almost 80% of patients experiencing no AEs during observation. Potentially treatment-related AEs were recorded, leading to permanent discontinuation of treatment in only 11% of cases (1.9% of the entire treated population). Psychiatric disorders, liver function marker elevation, gastrointestinal and skin disorders are the most prevalent AEs in our cohort, in line with the current literature. Special attention to long-term assessment in pediatric patients is a crucial point. Since a significant number of AEs occur after one year of treatment [21], longitudinal monitoring appears to be essential. National registries might provide a useful complement with a view to AE data collection, including occurrences related to mental health issues.

## Figures and Tables

**Figure 1 jcm-14-04335-f001:**
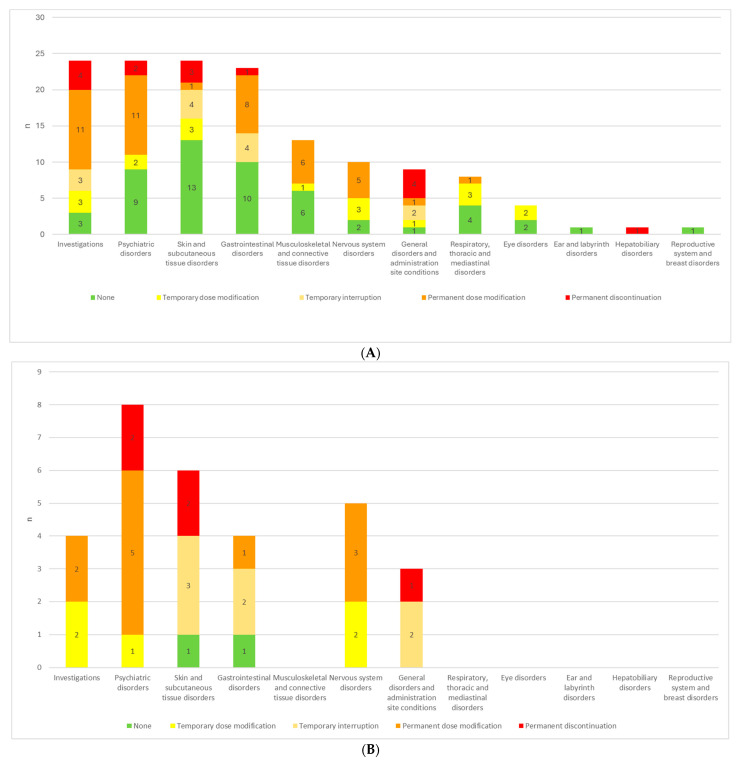
(**A**) ETI treatment adjustments in response to AEs for each SOC in all patients. (**B**) ETI treatment adjustments in response to AEs for each SOC in pediatric patients. (**C**) ETI treatment adjustments in response to AEs for each SOC in adult patients.

**Table 1 jcm-14-04335-t001:** Comparison of baseline characteristics for overall population, pwCF presenting at least 1 AE, and patients with (pw)CF not presenting AEs.

Variable	Overall Population	pwCF Presenting at Least 1 AE	pwCF Not Presenting AE	*p*-Value
	(N = 414)	(N = 85) (21%)	(N = 329)	
Age		85	329	0.4
Median (Q1, Q3)	27 (18, 38)	28 (18, 40)	27 (18, 37)	
Mean (SD)	28 (13)	29 (14)	28 (13)	
Age				0.7
Adult	310 (75%)	65 (21%)	245 (79%)	
Ped	104 (25%)	20 (19%)	84 (81%)	
Sex				0.6
F	222 (54%)	48 (22%)	174 (78%)	
M	192 (46%)	37 (19%)	155 (81%)	
Baseline ppFEV1 (N = 401)	401	81	320	0.2
Median (Q1,Q3)	77 (57, 92)	75 (56, 86)	77 (57, 93)	
Mean (SD)	74 (22)	71 (22)	75 (22)	
Baseline ppFEV1 (N = 401)				0.082
<40	31 (7.7%)	10 (32%)	21 (68%)	
≥40	370 (92%)	71 (19%)	299 (81%)	
Genotype				0.4
F508del/other	274 (66%)	53 (19%)	221 (81%)	
F508del/F508del	140 (34%)	32 (23%)	108 (77%)	
Cl at sweat test in mmol/L (N = 405)	405	85	320	0.5
Median (Q1, Q3)	96 (84, 105)	92 (84, 104)	96 (84, 106)	
Mean (SD)	93 (21)	93 (19)	93 (22)	

**Table 2 jcm-14-04335-t002:** Rate of AEs throughout the entire treatment period (Part A) and in the first 30 days alone (Part B). Comparison of rates in relation to the main patient characteristics.

		AE (n)	ETI Treatment Days (n)	AE/1000 Days ETI	*p*
**Part A**	Entire period of observation				
	Total	142	488,889	0.29	
Sex	M	64	224,985	0.284	0.9
	F	78	263,904	0.296	
Genotype	F508del/F508del	45	152,652	0.295	0.9
	F508del/other	97	336,237	0.288	
ppFEV_1_	ppFEV_1_ < 40	16	52,932	0.302	0.8
	ppFEV_1_ ≥ 40	118	415,512	0.284	
Sweat test	Cl < median (95.5)	76	232,524	0.327	0.3
	Cl ≥ median (95.5)	66	244,858	0.27	
Age	Age < 18 years	30	93,983	0.319	0.6
	Age ≥ 18 years	112	394,907	0.284	
**Part B**	0–30 days				
	Total	62	14,062	4.409	
Sex	M	24	6549	3.665	0.3
	F	38	7513	5.058	
Genotype	F508del/F508del	19	4590	4.139	0.8
	F508del/other	43	9472	4.54	
ppFEV_1_	ppFEV_1_ < 40	11	1110	9.91	0.01
	ppFEV_1_ ≥ 40	49	12,442	3.938	
Sweat test	Cl < median (95.5)	33	6772	4.873	0.5
	Cl ≥ median (95.5)	29	7020	4.131	
Age	Age < 18 years	15	3363	4.46	1
	Age ≥ 18 years	47	10,699	4.393	
**Part C**	0–15 days				
	Total	47	7521	6.249	
Sex	M	19	3497	5.433	0.5
	F	28	4024	6.958	
Genotype	F508del/F508del	14	2448	5.719	0.8
	F508del/other	33	5073	6.505	
ppFEV_1_	ppFEV_1_ < 40	11	592	18.581	0.001
	ppFEV_1_ ≥ 40	34	6657	5.107	
Sweat test	Cl < median (95.5)	27	3633	7.432	0.3
	Cl ≥ median (95.5)	20	3744	5.342	
Age	Age < 18 years	14	1809	7.739	0.4
	Age ≥ 18 years	33	5712	5.777	

**Table 3 jcm-14-04335-t003:** System Organ Class (SOC) for reported AEs.

SOC	AEs									
	Total		Age				Time *			
			Pediatric		Adult		0–15 days		>15 days	
	N = 142		N = 30		N = 112		N = 47		N = 87	
	n	(%)	n	(%)	n	(%)	n	(%)	n	(%)
**Investigations**	24	16.90	4	13.33	20	17.86	2	4.26	21	24.14
*Liver function marker elevations*	22		4		18		2		20	
*Pancreatic enzyme elevation*	1				1				1	
*Hypercholesterolemia*	1				1					
**Psychiatric disorders**	24	16.90	8	26.67	16	14.29	7	14.89	16	18.39
*Depression*	4		1		3		1		3	
*Low mood*	4				4		1		3	
*Brain fog*	3		1		2		1		2	
*Insomnia*	3				3		2		1	
*Behavioral alterations*	3		3				1		2	
*Attention problem*	2		2						2	
*Anxiety*	2				2		1		1	
*Anxiety symptoms*	1				1					
*Mood swing*	1				1				1	
*Agitation*	1		1						1	
**Skin and subcutaneous tissue disorders**	24	16.90	6	20.00	18	16.07	10	21.28	11	12.64
*Skin rash*	16		6		10		10		5	
*Itching*	3		0		3				2	
*Erythema*	2		0		2				2	
*Acne*	1				1					
*Mucosal dryness*	1				1				1	
*Desquamation of the palms*	1				1				1	
**Gastrointestinal disorders**	23	16.20	4	13.33	19	16.96	13	27.66	10	11.49
*Epigastric pain*	11		3		8		7		4	
*Diarrhea*	4				4		2		2	
*Abdominal pain*	2				2		2			
*Nausea*	2		1		1		1		1	
*Dyspeptic disorders*	2				2				2	
*Abdominal swelling*	1				1		1			
*Odontologic symptoms*	1				1				1	
**Musculoskeletal and connective tissue disorders**	13	9.15	-	-	13	11.61	2	4.26	10	11.49
*Myalgia*	6				6		2		4	
*Articular pain*	4				4				4	
*Muscle weakness*	2				2				1	
*Low back pain*	1				1				1	
**Nervous system disorders**	10	7.04	5	16.67	5	4.46	5	10.64	5	5.75
*Headache*	6		2		4		4		2	
*Drowsiness*	1				1				1	
*School difficulties*	1		1						1	
*Episodes of absence*	1		1				1			
*Tic*	1		1						1	
**General disorders and administration site conditions**	9	6.34	3	10.00	6	5.36	4	8.51	5	5.75
*Asthenia*	4		1		3		1		3	
*Fever*	2		1		1		1		1	
*Generalized edema*	1				1		1			
*Hyperpyrexia*	1		1				1			
*Tiredness*	1				1				1	
**Respiratory, thoracic and mediastinal disorders**	8	5.63	-	-	8	7.14	4	8.51	4	4.60
*Chest tightness*	3				3		2		1	
*Chest heaviness*	2				2				2	
*Blood in sputum*	2				2		2			
*Dysphonia*	1				1				1	
**Eye disorders**	4	2.82	-	-	4	3.57			2	2.30
*Dryness in the eyes*	2				2				1	
*Decrease in visual acuity*	1				1					
*Eyelid swelling*	1				1				1	
**Ear and labyrinth disorders**	1	0.70			1	0.89			1	1.15
**Hepatobiliary disorders**	1	0.70			1	0.89			1	1.15
**Reproductive system and breast disorders**	1	0.70			1	0.89			1	1.15

* date of occurrence not available in 8 cases.

**Table 4 jcm-14-04335-t004:** Modifications of ETI therapy in response to AEs, broken down by SOC. The frequency of the worst outcome is reported for each SOC.

Permanent	Permanent Dosage Modification	Temporary Interruption	Temporary Dosage Modification	None	SOC
Discontinuation					
4/24 (16.7%)	11/24 (45.8%)	3/24 (12.5%)	3/24 (12.5%)	3/24 (12.5%)	Investigations
2/24 (8.3%)	11/24 (45.8%)		2/24 (8.3%)	9/24 (37.5%)	Psychiatric disorders
3/24 (12.5%)	1/24 (4.2%)	4/24 (16.7%)	3/24 (12.5%)	13/24 (54.2%)	Skin and subcutaneous tissue disorders
1/23 (4.3%)	8/23 (34.8%)	4/23 (17.4%)		10/23 (43.5%)	Gastrointestinal disorders
	6/13 (46.2%)		1/13 (7.7%)	6/13 (46.2%)	Musculoskeletal and
					connective tissue disorders
	5/10 (50.0%)		3/10 (30.0%)	2/10 (20.0%)	Nervous system disorders
4/9 (44.4%)	1/9 (11.1%)	2/9 (22.2%)	1/9 (11.1%)	1/9 (11.1%)	General disorders and
					administration site conditions
	1/8 (12.5%)		3/8 (37.5%)	4/8 (50.0%)	Respiratory, thoracic and
					mediastinal disorders
			2/4 (50.0%)	2/4 (50.0%)	Eye disorders
				1/1 (100%)	Ear and labyrinth disorders
1/1 (100%)					Hepatobiliary disorders
				1/1 (100%)	Reproductive system and breast disorders

## Data Availability

The raw data supporting the conclusions of this article will be made available by the authors on request.

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
