# Peer review of "Reported Adverse Events in Patients with CF Receiving Treatment with Elexacaftor/Tezacaftor/Ivacaftor: 5 Years Observational Study"

_jcm, 2025, doi:10.3390/jcm14124335_

Round 1
Reviewer 1 Report
Comments and Suggestions for Authors
The Authors describe adverse events in CF patients treated with ETI for 5 years. The information emerging from this study enriches the current literature on the topic. However several critical points of the manuscript should be addressed:
- line 12: please change "patients" with "people";
- lines 13, 37: the name of the gene should be reported in italics throughout the manuscript;
- lines 90-100: was the normality of the data distributions assessed? If so, with which test?
- Table 1: the mean (standard deviation) should be reported for continuous parametric (normal) data or the median (interquartile range) for non-parametric data. Furthermore, the choice of the statistical test (T test or Mann Whitney) should be made based on the normality or not of the data distributions. It is therefore recommended to review the statistical analysis of all data;
- Table 1: it would be appropriate to report in the table the p values ​​of the comparisons between adults and children, males and females, ppFEV1 < 40 vs ppFEV1 > 40, F508del homozygous vs heterozygous;
- lines 124-126: Table 2 is not clearly explained in the text, it is not clear what the percentages reported in the text correspond to;
- line 132: "followed by skin and subcutaneous tissue disorders.." is not entirely accurate since the number of AEs is the same as that of Investigations and psychiatric disorders, please correct;
- Table 3: I would recommend reporting the AEs in adults and children separately in two columns with statistical analysis;
- Table 4: it is redundant with respect to Figure 1; I would recommend separating adults from children here too;
- results and discussion should be revised based on the review of the statistical analysis.
Author Response
The Authors describe adverse events in CF patients treated with ETI for 5 years. The information emerging from this study enriches the current literature on the topic. However several critical points of the manuscript should be addressed:
- line 12: please change "patients" with "people";
R1.1. changed in manuscript
lines 13, 37: the name of the gene should be reported in italics throughout the manuscript;
R1.2. CFTR gene has been reported in italics, as recommended, whereas CFTR protein in normal font.
lines 90-100: was the normality of the data distributions assessed? If so, with which test?
R1.3. The normality of the distribution was assessed by using the Shapiro-Wilk Test. Methods section has been improved accordingly (lines 98-101).
Table 1: the mean (standard deviation) should be reported for continuous parametric (normal) data or the median (interquartile range) for non-parametric data. Furthermore, the choice of the statistical test (T test or Mann Whitney) should be made based on the normality or not of the data distributions. It is therefore recommended to review the statistical analysis of all data;
R1.4. Table 1 was modified to include median and Q1, Q3 values. In the statistical section, the following sentence was included: “Chi-square was used for categorical variables and, after having assessed the normality of the distribution for the continuous variables by the Shapiro-Wilk Test, the Mann-Whitney test was used, because the normality was not verified” (lines 98-101).
Table 1: it would be appropriate to report in the table the p values ​​of the comparisons between adults and children, males and females, ppFEV1 < 40 vs ppFEV1 > 40, F508del homozygous vs heterozygous;
R1.5. The p-values in Table 1 indicate the association between the presence of at least one AE and various patient characteristics, whereas Table 2 examines differences in patient characteristics across reaction types.
lines 124-126: Table 2 is not clearly explained in the text, it is not clear what the percentages reported in the text correspond to;
R1.6. To clarify how the percentages were calculated, the manuscript was revised by adding lines 122–125, removing the reference to Table 2 in that section.
line 132: "followed by skin and subcutaneous tissue disorders.." is not entirely accurate since the number of AEs is the same as that of Investigations and psychiatric disorders, please correct;
R1.7. Text has been edited accordingly (lines 140-147)
Table 3: I would recommend reporting the AEs in adults and children separately in two columns with statistical analysis;
R1.8. table 3 was modified in the revised manuscript. A comment on this table was added (lines 140-145, 276-280)
Table 4: it is redundant with respect to Figure 1; I would recommend separating adults from children here too;
R.1.9. Figure 1 has been modified by separating children and adults. A description of this table was included in the revised manuscript (lines 184-189). Table 4 was included in the revised version as supplementary material (new Table S1)
results and discussion should be revised based on the review of the statistical analysis.
R1.10. Results and discussion have been improved and revised accordingly.

Reviewer 2 Report
Comments and Suggestions for Authors
the authors provide the results of a single center prospective study that aimed to examine and describe a cohort of CF patients receiving ETI therapy and the associated side effects the individuals experienced. this is a timely study providing early long term data on this topic of interest to me as a CF physician. although the authors do not necessarily comment on this - my experience and the experience i hear from other CF providers is that the side effect profile of ETI in real world use differs from that of prior clinical trials and side effects are common and sometimes not necessarily discussed between providers and patients during the first several years of ETI- there are several potential reasons for this and the authors should include this in introduction including patients not recognizing side effect and confusing it for normal aging, patients hesitant to discuss side effects when overall the drug has significantly improved their quality of life etcetera- this should be elaborated on by the authors and they should cite similar examples of prior studies regarding real world side effect profile of ETI. Next- the authors do not describe their methodology suffiently- specifcally how the side effects were identified- did patients do a questionairre? did the provider ask specific questions? was it only if it came up by the patient etctera. this is a key weakness of the study and without greater detail on this the study is not yet publishable- depending on the methodology the authors next need to also describe potential weaknesses of the method.
the data provided is very helpful but can be improved - specifically the side effect profiles should have a chart split by peds/adult/all - although they provide some specific data points in the discussion regarding differences i didnt see a table that addressed this and may identify differences in side effect profiles across the age spectrum/lifespan. in addition the data section includes sweat chloride- the data provided are quite high- were these sweats on ETI or w/o a modulator? they seem unexpectedly high if it is on ETI. The data presented did not describe if a side effect improved using a dose modification and if available this data should be provided. along those lines- attention deficit disorder is a disease that requires neurocognitive testing for the diagnosis- similar can be said about the other side effects (ie depression/anxiety)- did these happen or is this just reported symptom? if it is just a reported symptom which i think is the case then the author needs to delineate that in the methods section as well. also with regards to side effects- can the authors separate early side effects w/ treatment and late side effects- this has significant clinical importance- it seems to me rash for example typically occurs in first 2 weeks where as the neurocognitive side effects is typically after several weeks. it would be helpful if the authors could provide their data regarding this.
Author Response
The authors provide the results of a single center prospective study that aimed to examine and describe a cohort of CF patients receiving ETI therapy and the associated side effects the individuals experienced. this is a timely study providing early long term data on this topic of interest to me as a CF physician. although the authors do not necessarily comment on this - my experience and the experience i hear from other CF providers is that the side effect profile of ETI in real world use differs from that of prior clinical trials and side effects are common and sometimes not necessarily discussed between providers and patients during the first several years of ETI- there are several potential reasons for this and the authors should include this in introduction including patients not recognizing side effect and confusing it for normal aging, patients hesitant to discuss side effects when overall the drug has significantly improved their quality of life etcetera- this should be elaborated on by the authors and they should cite similar examples of prior studies regarding real world side effect profile of ETI.
R2.1. We thank the reviewer for her/his valuable feedback. We have revised the discussion in accordance with reviewer’s recommendations, highlighting that this aspect of real-life studies, and of our methods in particular, also constitutes a limitation of the study.
Next- the authors do not describe their methodology suffiently- specifcally how the side effects were identified- did patients do a questionairre? did the provider ask specific questions? was it only if it came up by the patient etctera. this is a key weakness of the study and without greater detail on this the study is not yet publishable- depending on the methodology the authors next need to also describe potential weaknesses of the method.
R2.2. AEs were collected actively by physicians during outpatient and inpatient visits, alongside chart and records review during observation period. As an internal protocol when suspect arose of psychological disorder, or when pwCF reported psychological symptoms, an interview with CF psychologist was organized and if deemed necessary, patient was sent to child neuropsychiatric specialist or psychiatric specialist. Method session was modified (lines 78-83). This of course represent a limit which we now have added in discussion session (lines 200-204).
the data provided is very helpful but can be improved - specifically the side effect profiles should have a chart split by peds/adult/all - although they provide some specific data points in the discussion regarding differences i didnt see a table that addressed this and may identify differences in side effect profiles across the age spectrum/lifespan.
R2.3. Table 3 was modified in the revised manuscript, and a comment on this table was added (lines 140-147). Figure 1 was modified to better characterise adults and pediatric AEs.
in addition the data section includes sweat chloride- the data provided are quite high- were these sweats on ETI or w/o a modulator? they seem unexpectedly high if it is on ETI.
R2.4. Those reported in table 1 are baseline values before ETI initiation
The data presented did not describe if a side effect improved using a dose modification and if available this data should be provided.
R2.5. Reducing the ETI dose enabled continuation of treatment while preserving its clinical efficacy. Figure 1 illustrates the number of patients who required permanent discontinuation versus those who needed dosage adjustments. This approach allowed a greater proportion of patients to remain on therapy despite AEs, without compromising clinical efficacy. Text has been integrated with these information.
along those lines- attention deficit disorder is a disease that requires neurocognitive testing for the diagnosis- similar can be said about the other side effects (ie depression/anxiety)- did these happen or is this just reported symptom? if it is just a reported symptom which i think is the case then the author needs to delineate that in the methods section as well.
R2.6: In the manuscript, we replaced “deficit” with “concentration problem,” and this change has been reflected in both the tables and the discussion.
also with regards to side effects- can the authors separate early side effects w/ treatment and late side effects- this has significant clinical importance- it seems to me rash for example typically occurs in first 2 weeks where as the neurocognitive side effects is typically after several weeks. it would be helpful if the authors could provide their data regarding this.
R2.7. Table 3 was modified to include AEs occurrence in the first 2 weeks and afterwards. We discussed results throughout the text (lines 148-151, 262-268).

Reviewer 3 Report
Comments and Suggestions for Authors
This is an important manuscript as it presents long term data on a substantial population with CF on ETI. It is well written and the figures and tables are well presented
I have only minor clarifications
- How were the neuropsychiatric side effects quantified? Was this assessed prospectively? What tools of assessment were used and at which age group? Were medications commenced to manage these AEs? Was a psychiatrist/ psychologist involved in the care of the patients? What was the algorithm used for dose modification?
2. Can details of the eye disturbances be presented in greater detail?
3. details of patients usual therapies would be useful. Specifically, how many were in, alpha Dornase, macrolides, URSO?
Author Response
This is an important manuscript as it presents long term data on a substantial population with CF on ETI. It is well written and the figures and tables are well presented.
R.3.1. We are grateful to this reviewer for her/his kind appreciation in our study.
I have only minor clarifications
How were the neuropsychiatric side effects quantified? Was this assessed prospectively? What tools of assessment were used and at which age group? Were medications commenced to manage these AEs? Was a psychiatrist/ psychologist involved in the care of the patients?
R3.2: Adverse events were actively monitored by physicians during both outpatient and inpatient visits and via chart and record reviews throughout the observation period. Per our internal protocol, if a psychological disorder was suspected or patients with CF reported psychological symptoms, an interview with a CF psychologist was arranged. If deemed necessary, the patient was then referred to a child neuropsychiatric or psychiatric specialist. This has been clarified in method section.
For low mood or anxiety symptoms, we counted any symptom reported by the patient, whereas depression or anxiety disorders were recorded only if a diagnosis was confirmed by a neuropsychiatric or psychiatric specialist. Table 3 has been revised to clarify this distinction.
In our cohort, psychiatric medications were prescribed to three patients following a new diagnosis by a psychiatric specialist, and treatment was adjusted by a specialist for three additional patients who experienced symptom worsening during ETI therapy.
What was the algorithm used for dose modification?
R.3.3. An internal protocol was applied. The manuscript has been updated (lines 162-169), and supplementary data has been added to provide a detailed description of the protocol.
Can details of the eye disturbances be presented in greater detail?
R.3.4. Better definition was integrated in the manuscript in Table 3
- details of patients usual therapies would be useful. Specifically, how many were in, alpha Dornase, macrolides, URSO?
R.3.5. Each time ETI was prescribed for the first time, concomitant therapies were reviewed for potential interactions. Patients were informed of the importance of adhering to their regular treatment regimens, and physicians exercised caution when adjusting maintenance therapies. The use of DNase, macrolides, and ursodeoxycholic acid was not included in baseline characteristics, as our study focused on adverse events rather than efficacy, which could be confounded by these therapies.

Round 2
Reviewer 1 Report
Comments and Suggestions for Authors
The Author have addressed all comments/suggestions.
Reviewer 2 Report
Comments and Suggestions for Authors
authors sufficiently addressed all of my prior edits- well done i think it is ready for publication